# Zirconium and hafnium catalyzed C−C single bond hydroboration

Sida Li[1,2], Haijun Jiao [3] ✉, Xing-Zhong Shu [4] & Lipeng Wu [1,5] ✉

Selective cleavage and subsequent functionalization of C−C single bonds present a fundamental challenge in synthetic organic chemistry. Traditionally, the activation of C−C single bonds has been achieved using stoichiometric transition-metal complexes. Recently, examples of catalytic processes were developed in which use is made of precious metals. However, the use of inexpensive and Earth-abundant group IV metals for catalytic C−C single-bond cleavage is largely underdeveloped. Herein, the zirconium-catalyzed C−C single-bond cleavage and subsequent hydroboration reactions is realized using $Cp_2ZrCl_2$ as a catalytic system. A series of structures of various γ-boronated amines are readily obtained, which are otherwise difficult to obtain. Mechanistic studies disclose the formation of a N–$Zr^{IV}$ species, and then a β-carbon elimination route is responsible for C−C single bond activation. Besides zirconium, hafnium exhibits a similar performance for this transformation.

Selective cleavage and subsequent functionalization of C−C single bonds present a fundamental challenge in catalysis and synthesis[1–11]. This is mainly due to the relatively high bonding energy (BE, about 355 kJ/mol) and directional σ orbitals of C−C bonds. In addition, the competitive C−H bond activation (about 400 kJ/mol, but statistically abundant) also causes chemoselectivity problems[12–22]. Nevertheless, cleavage and functionalization of C−C single bonds are attracting increasing attention in synthetic organic chemistry because it offers a unique and straight route to target molecules/structures. Synthetic chemists have developed various strategies for the activation of C−C single bonds. They are mainly classified into two mechanistic categories: oxidative addition and β-carbon elimination, associated with metal centers (Fig. 1a). Besides the use of stoichiometric transition-metal complexes[23–26], examples of catalytic processes have been reported in recent years−most involve the use of precious metals[27–45].

Early-transition metals have different electron configurations from late ones. Thus, their complexes often show other or orthogonal reactivities with late-transition metal complexes[46–61]. In addition, they are also Earth-abundant (e.g., zirconium is almost as abundant as carbon in the Earth's upper continental crust). However, zirconium usually exists in the $Zr^{IV}$ oxidation state, which is not viable for direct oxidative addition activation of a single chemical bond. Thus, in situ generated or isolable low-valent zirconocene complexes ($Zr^{II}$), such as Negishi's $(Cp_2ZrBu_2)$[62] and Rosenthal's reagents $(Cp_2Zr(py) Me_3SiC≡CSiMe_3)$[63], for the activation of B−H[64,65], Si−H[66–69] bonds were reported. C−C single bond cleavage by a zirconium species has also been reported intermittently since the 1990s[70–73]. In 1994, Rosenthal described the activation of conjugated C−C single bonds of a 1,3-butadiyne moiety (C≡C−C≡C) using Rosenthal's reagent, resulting in a dimeric complex (Fig. 1b)[74]. Dimmock and Whitby also found that zirconocene η²-alkene and η²-imine complexes with adjacent cyclopropane rings could undergo cyclopropane ring cleavage[75]. Then, in 2014, Marek reported an expedient approach, including allylic C−H activations followed by C−C single bond activation (Fig. 1c)[76]. It is also worth mentioning that since the 1990s, Negishi, Takahashi, and Xi have studied the chemistry of zirconacycles, the transformation of which with other unsaturated molecules usually involved a β, β'-C−C bond cleavage[77–81]. All the former instances used (over) stoichiometric amounts of zirconium, and no precedents of catalytic methods using homogeneous zirconium catalysis had been developed−to the best of

[1]State Key Laboratory for Oxo Synthesis and Selective Oxidation, Lanzhou Institute of Chemical Physics (LICP), Chinese Academy of Sciences, Lanzhou 730000, PR China. [2]University of Chinese Academy of Sciences, Beijing 100049, PR China. [3]Leibniz-Institut für Katalyse e. V., Albert-Einstein-Straße 29a, 18059 Rostock, Germany. [4]State Key Laboratory of Applied Organic Chemistry, College of Chemistry and Chemical Engineering, Lanzhou University, Lanzhou 730000, PR China. [5]College of Material Chemistry and Chemical Engineering, Key Laboratory of Organosilicon Chemistry and Material Technology, Ministry of Education, Hangzhou Normal Univerisity, Hangzhou 311121, PR China. ✉e-mail: haijun.jiao@catalysis.de; lipengwu@licp.cas.cn

**Fig. 1 | Strategies for C−C single bond activation. a** Transition metal mediated C−C single bond activation; **b, c** stoichiometric amount of zirconium-mediated C−C single bond activation; **d** zirconium and hafnium catalyzed C−C single bond activation.

our knowledge (The use of heterogeneous zirconium catalysis for C−C bond cleavage was reported by Basset[82–84]). Consequently, activating C−C single bonds with zirconium catalysis for chemical transformation remains a significant challenge. It is of considerable scientific and practical interest to synthetic organic chemistry to address this. Herein, we report the development of an unprecedented catalytic system that resulted in the realization of the zirconium- and hafnium-catalyzed C−C single-bond hydroboration (Fig. 1d). Mechanistic studies support the formation of N−Zr$^{IV}$ species and then a β-carbon elimination route for C−C single bond activation. Our work provides an alternative catalytic method for C−C single bonds hydroboration, and establishes the bond activation models and catalytic application of group IV transition metals.

## Results

### Catalytic reaction investigations

As a synthetically significant transformation in organic chemistry, hydroboration of C=C bonds is well-studied. However, catalytic hydroboration of C−C single bonds remains underdeveloped. Only two systems using Ir and Rh are known for the hydroboration of cyclopropanes, as developed by Yamaguchi[85] and Shi[86,87]. Besides making use of precious metals and *N*- or *P*-ligands, it is noticed that for

the Ir system, the careful choice of a chiral *t*Bu-Quinox ligand is crucial for the C−C bond hydroboration over the C−H boration[88–90]. For the Rh system, the PPh₃ ligand is essential in inhibiting side reactions such as the formation of alkenes. Furthermore, in the former case, cleavage of $C_β$−$C_{β'}$ bond is observed, while the latter cleavages $C_α$−$C_β$ bond. Thus, it is still highly desirable to develop a facile and inexpensive catalytic system for the hydroboration of C−C single bonds.

We commenced our investigation using 0.2 mmol of *N*-Piv-cyclo-propylamines (**1a**) with 1.5 equiv. pinacolborane (HBpin) in 1 mL toluene at 120 °C as the model reaction, using 5 mol% Cp₂ZrCl₂ as catalyst (Table 1). Our preliminary investigations unveiled that the addition of 1 equiv. of base is the key for the zirconium-catalyzed C−C bond hydroboration (Supplementary Table 1); K₂CO₃ was the optimal choice (Table 1, entry 1, 81% yield of **2a**). Various other zirconium complexes were then tested but were unsuccessful. There was no reaction with the sterically bulkier Cp*₂ZrCl₂ (Table 1, entry 2). With Cp₂ZrHCl as catalyst, a 50% yield of **2a** was obtained, whereas with Cp₂ZrMe₂ only 10% **2a** was produced (Table 1, entries 3 and 4). It was evident that without Cp₂ZrCl₂ or K₂CO₃ no reaction proceeded (Table 1, entries 5 and 6). Interestingly, using Cp₂TiCl₂ instead of Cp₂ZrCl₂ also gave no reaction (Table 1, entry 7). An attempt was then made to reduce the amount of K₂CO₃. Pleasingly, even with 0.1 equiv.

**Table 1 | Zr-catalyzed hydroboration of cyclopropylamines—condition optimization[a]**

| Entry | Catal. | x | 2a yield (%)[b] |
|---|---|---|---|
| 1 | Cp₂ZrCl₂ | 1 | 81 |
| 2 | Cp*₂ZrCl₂ | 1 | - |
| 3 | Cp₂ZrHCl | 1 | 50 |
| 4 | Cp₂ZrMe₂ | 1 | 10 |
| 5 | Cp₂ZrCl₂ | 0 | - |
| 6 | - | 1 | - |
| 7 | Cp₂TiCl₂ | 1 | - |
| 8 | Cp₂ZrCl₂ | 0.1 | 68 |
| **9** | **Cp₂ZrCl₂** | **0.3** | **91** |
| 10 | Cp₂ZrH₂ | 0 | 95 |

$^a$Reaction conditions: 0.2 mmol **1a**, HBpin (1.5 equiv.), catalyst (5 mol%), $K_2CO_3$ (0.1–1 equiv.), and 1 mL toluene in a 15 mL pressure tube at 120 °C for 24 h.
$^b$Yields were determined by GC using dodecane as an internal standard.

$K_2CO_3$ already had a 68% yield of **2a** (Table 1, entry 8). Finally, a much higher yield of **2a** (91%) was obtained with just 0.3 equiv. $K_2CO_3$ (Table 1, entry 9). In addition, 95% yield of **2a** was obtained using 5 mol% $Cp_2ZrH_2$ as catalyst without $K_2CO_3$ (Table 1, entry 10). However, considering the simplicity of using readily available and inexpensive $Cp_2ZrCl_2$ as the catalyst, we conducted the following studies using $Cp_2ZrCl_2/K_2CO_3$ system (Table 1, entry 9). The results with $Cp_2ZrH_2$ gave us some clues for the subsequent mechanism studies (vide infra).

### Substrates scope studies

Having the reaction conditions for the Zr-catalyzed hydroboration of cyclopropylamines in hand (Table 1, entry 9), various cyclopropane rings were investigated to establish the generality of our methodology (Fig. 2).

First, the tolerance of substituents on the *para*-position of the phenyl ring was studied. We found that electron-neutral, electron-donating, and electron-withdrawing groups are tolerated; moderate to good yields were obtained (**2a-2k**, 40-82% yields). In general, electron-donating groups (−Me, −OMe, −$^t$Bu, −SMe, **2c−2e, 2k**) gave better results than electron-withdrawing groups such as −CF₃ (**2f**). Halide substituents −F, −Cl, −Br, which likely undergo competitive hydro-dehalogenation or boration reactions, are untouched in our system (**2g−2i**). We found that steric effects have some influence on the results, as changing the substituents from the *para*-position to the *meta*- and *ortho*-position, led to slightly decreased yields or the need for higher reaction temperatures (**2l−2o**). Significantly, naphthyl, benzodioxole, alkyne, and heteroaromatic rings such as furyl and thiophene substituents are all compatible in our system, with yields in the range of 71 − 78% (**2p−2t**). Similarly, good results were obtained when the phenyl group is at the β′ position (**2u**). Changing R¹ from an aryl to an alkyl group was also successful, with both acyclic and cyclic alkyl substituents (up to 77% yield, **2v−2ac**). Pleasingly, products **2q, 2ab** and **2ac** were obtained in 69%, 77%, and 67% yields, respectively, with no double or triple bond interference. Moreover, cyclopropane rings with two substituents on the R¹ and R² positions were also suitable (**2ad−2ae**).

The effect of the substituent on the amide groups was then studied. When R is 1-methylcyclohexyl, the hydroboration product **2af** was obtained in 76% yield. Changing R to a sterically bulkier adamantly group resulted in a slightly lower yield (**2ag**, 56%). Substrates with 2,2-dimethylbutyl and 1-methylcyclopropyl, and substituents containing chloride are all converted to their corresponding hydroboration

products **2ah−2aj** in yields of up to 64%. Sulfonamide is also tolerated in our system, which get the hydroboration product **2ak** in 66% yield. The reaction also proceeded with thioamide (**2al**). Finally, we found that cyclopropylamines derived from *Oleanolic Acid* and *Gemfibrozil* also reacted well in our system; the corresponding products, **2am** and **2an**, were obtained in yields of 49% and 59%, respectively.

### Hafnium-catalyzed reaction

Compared with zirconium, hafnium has received less attention as a homogeneous catalyst in organic reactions. To our knowledge, reactivity toward C−C single bonds activation is also unknown. After successfully establishing zirconium-catalyzed C−C single bond activation of cyclopropylamines and their subsequent hydroboration, we further explored the reactivity of a hafnium complex towards C−C single bonds. We established that the base plays an essential role in tuning the reactivity. Eventually, $Cs_2CO_3$ was found to be the optimal base (Supplementary Table 5). Then, under the optimal reaction conditions, we conducted substrate scope generality studies (Fig. 3). We found that the hafnium system is not only suitable for substrates that work in the zirconium system but also for substrates that do not work there. For example, substrates with a −CN group do not react with the zirconium catalyst, but a 50% yield of product **2ao** was obtained with hafnium. Product **2ap**, with two fluorides on the phenyl ring, was also obtained in 51% yield. Additionally, cyclohexyl- (**2aq**) and phenyl-substituted substrates (**2ar**) were also applicable in the hafnium system.

### Synthetic derivation

The practical utilization of our system was then demonstrated on a gram scale (Fig. 4). When we subjected 10 mmol of **1a** to our standard reaction conditions, we obtained **2a** in 73% yield (1.97 g). Furthermore, the synthetic derivatization of **2a** was demonstrated. Using the aminoazanium of DABCO as an amination reagent[91], and then protecting the amine with TFAA, the corresponding TFA-amide **3a** was obtained in 66% yield. γ-Boronated amine **3b**, which is otherwise difficult to obtain[92], was obtained in 72% yield by reducing the amide functional group to an amine and then protecting it with TsCl. Product **2a** can be transformed into potassium trifluoroborate salt **3c** using KHF₂ (82% yield). Treating **2a** with furan-2-yllithium followed by NBS afforded the arylated product **3d** in 74% yield. Finally, Pd-catalyzed Suzuki−Miyaura coupling of **2a** with Estrone-derived triflate gave **3e** in 45% yield.

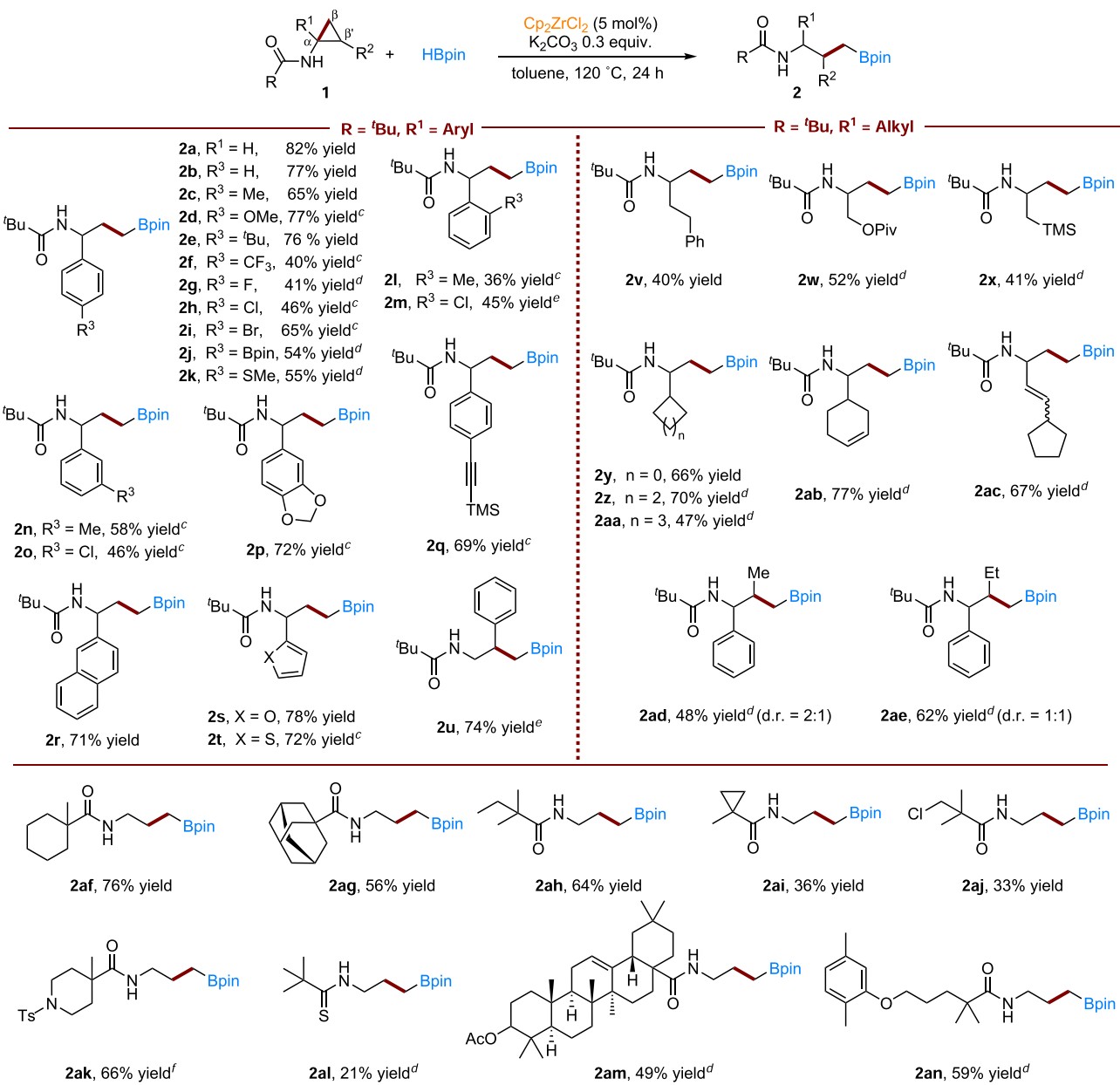

**Fig. 2 | Zirconium-catalyzed hydroboration of cyclopropylamines–generality studies.** [a,b] [a]Reaction conditions: 0.2 mmol **1**, HBpin (1.5 equiv.), K₂CO₃ (30 mol%), Cp₂ZrCl₂ (5 mol%) and 1 mL toluene in a 15 mL pressure tube at 120 °C for 24 h; [b]Isolated yields are given; [c]130 °C for 24 h; [d]150 °C for 24 h; [e]HBpin (2.0 equiv.), K₂CO₃ (60 mol%), Cp₂ZrCl₂ (10 mol%) at 150 °C for 24 h; [f]HBpin (2.0 equiv.), Cp₂ZrH₂ (5 mol%) at 150 °C for 24 h.

## Mechanistic studies

To shed light on the reaction mechanism, several control experiments were performed (Fig. 5). The possible formation of an alkene intermediate via ring-opening of cyclopropanes followed by hydroboration was studied. However, no alkenes were detected after 3 or 12 h under standard reaction conditions with or without HBpin (Fig. 5Aa, Supplementary Fig. 1). Utilization of alkenes **1a'** and **1a''** afforded less than 6% **2a** (Fig. 5Ab). When enantioenriched substrate (1*S*, 2*R*)-**1as** was applied, the desired product (*R*)-**2as** was obtained without erosion of the enantioselectivities (Fig. 5Ac, Supplementary Figs. 2, 3). Those results excluded a consecutive cyclopropane ring opening-hydroboration process. Then, the possibility of a reaction pathway that involved a radical species was investigated. TEMPO (2,2,6,6-tetramethylpiperidinyloxyl) (1−2 equiv) had almost no effect on the results. However, upon increasing the amount thereof (4 equiv.) the yields of **2a** decreased to 44% (Fig. 5B). At this point, it should be borne

in mind that TEMPO inhibition experiments can sometimes provide ambiguous results[93]. Thus, additional experiments with the addition of 9,10-dihydroanthracene (DHA) were conducted; no effect on the yield of **2a** was detected (Fig. 5B). The results with TEMPO and DHA excluded a radical mechanism.

Then, the active zirconium catalytic species was studied. Upon the combination of Cp₂ZrCl₂ and K₂CO₃ in d[8]-Tol heated at 120 °C for 12 h, a new species appeared around 6.0 ppm in the ¹H NMR spectrum (Fig. 5Ca). With 2 equiv. K₂CO₃ and heating for a longer reaction time, the Cp₂ZrCl₂ was fully converted to this new species (Fig. 5Cb). Then, the isolated new species was characterized by IR spectroscopy and was currently assigned to Cp₂ZrCO₃ by comparation with literature data (Supplementary Fig. 4)[94,95]. Nevertheless, upon further adding HBpin to the above solution, we could detect the formation of Zr−H species in the ¹H NMR spectrum by trapping with acetone (Fig. 5Cc, Supplementary Fig. 5). This finding, together with the fact that Cp₂ZrHCl or Cp₂ZrH₂ can

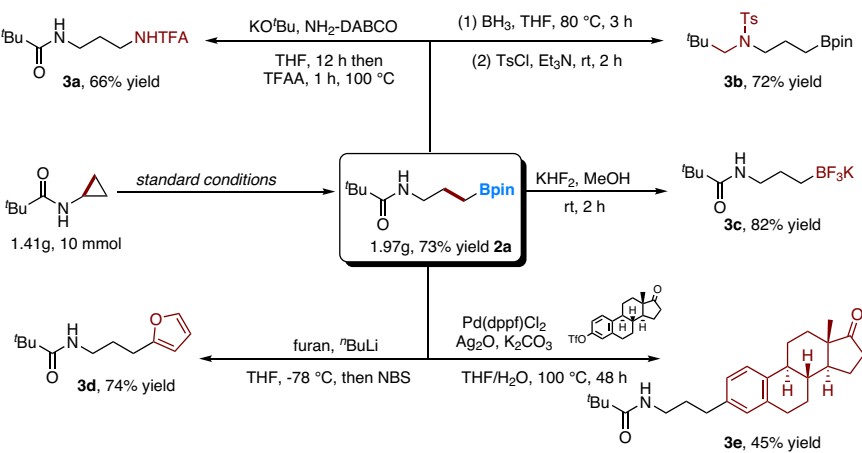

**Fig. 3 | Hafnium-catalyzed hydroboration of cyclopropylamines—generality studies.** [a,b,c] [a]Reaction conditions: 0.2 mmol **1**, HBpin (2.0 equiv.), Cs$_2$CO$_3$ (30 mol%), Cp$_2$HfCl$_2$ (5 mol%) and 1 mL toluene in a 15 mL pressure tube at 150 °C for 24 h; [b]HBpin (3.0 equiv.), Cs$_2$CO$_3$ (60 mol%), Cp$_2$HfCl$_2$ (10 mol%) and 1 mL toluene in a 15 mL pressure tube at 150 °C for 72 h; [c]Isolated yields are given.

**Fig. 4 | Synthetic Applications.** Reaction in gram scale and further derivatization of **2a**. Isolated yields are given; for detailed reaction conditions, please refer to the supplementary information.

catalyze the C−C bond hydroboration process without K$_2$CO$_3$ (65% and 95% yields, Supplementary Table 2), we concluded that Zr−H species are essentially the active catalysts via the consecutive reactions of Cp$_2$ZrCl$_2$, K$_2$CO$_3$, and HBpin (Fig. 5Cd). According to the work from Ganem[96], Rosenthal[97], and Cantat[98], the active Zr−H species can interact with **1a** to form N−Zr species via metathesis with N−H bond. This is further proved in our case that Cp$_2$ZrHCl reacts with the N−H group of **1a** with the release of H$_2$ or HD when **1a**-D was used (Fig. 5D, Supplementary Fig. 6). To add further proof of the importance of the N−H, *N*-methylated analog substrate **1a-Me**, and replace the N−H with CH$_2$ or O substrates **1a-C**, **1a-O** were subjected to our reaction conditions. As expected, no corresponding C−C bond hydroboration product were observed (Fig. 5E).

Keep in mind that β-carbon elimination is one of the main pathway for C−C bond cleavage. It is natural to think that after the formation of N−Zr species, a β-carbon elimination may proceed to cleavage the C−C bond to produce an imino propyl zirconium species. This is consistent with the fact that we can observe the presence of the putative imine intermediate both on GC/MS and $^{13}$C NMR when substrate **1u** was used (Fig. 5F, Supplementary Figs. 7, 8).

Based on the above mechanistic study and our DFT calculation results (Supplementary Fig. 9), we conclude the following general reaction pathway for our Cp$_2$ZrCl$_2$/K$_2$CO$_3$ system (Fig. 6). First, the in-situ formed Zr−H species reacts with N−H bonds of the substrates to form N−Zr$^{IV}$ species **A** via H$_2$ release (i). Next, the C−C single bond is

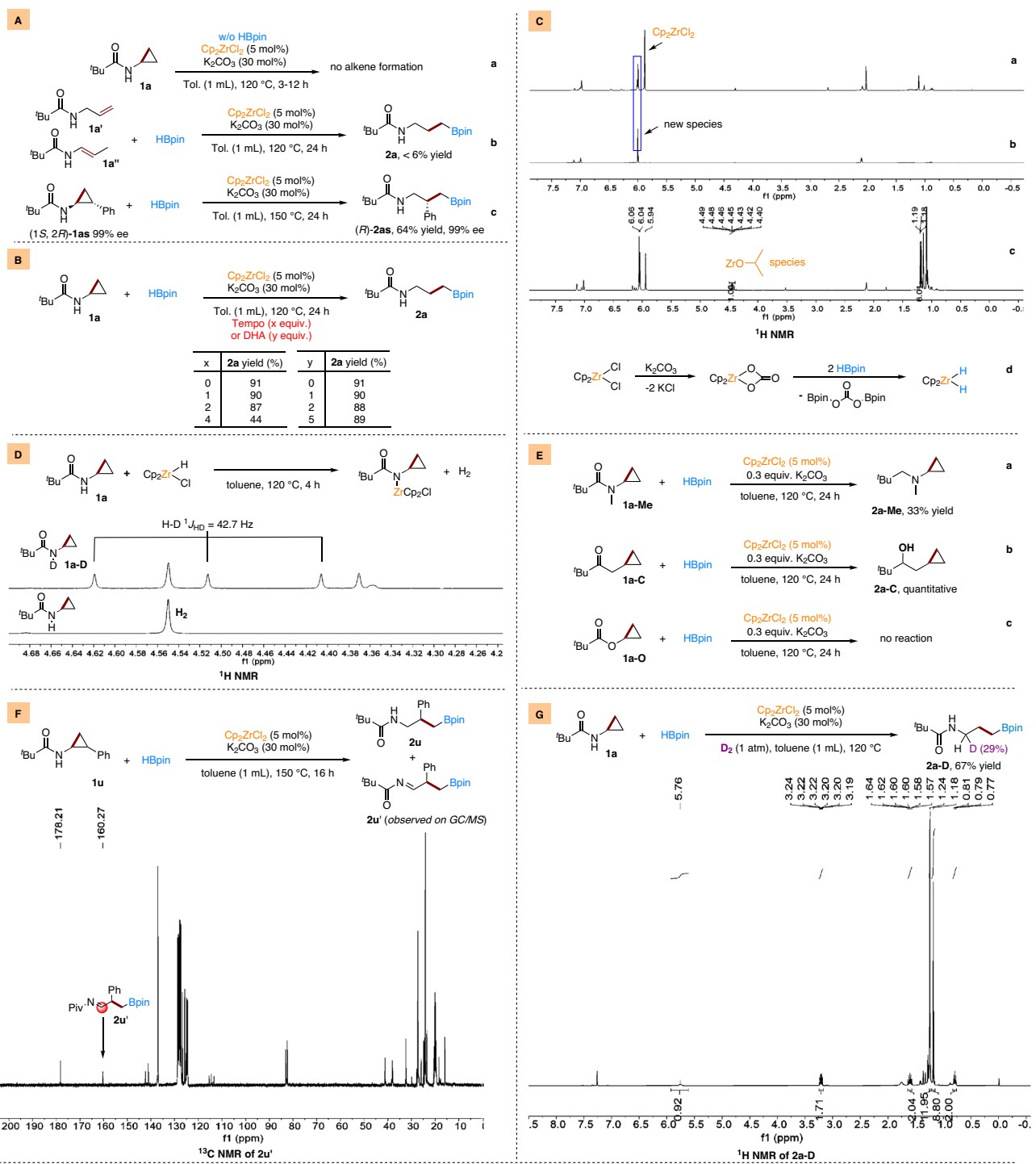

**Fig. 5 | Experimental mechanism studies. A** possible alkene intermediate formation; **B** possible radical pathway; **C** ¹H NMR spectra show the formation of Zr−H species; **D** ¹H NMR spectra show the release of H₂ or HD by reacting Cp₂ZrHCl with **1a**; **E** Control experiments to show the importance of N−H bond; **F** Detection of imide intermediate **2u'** by crude ¹³C NMR spectroscopy; **G** Deuterium labeling experiment by introducing D₂ in the standard reaction.

cleaved via β-carbon elimination of intermediate **A** to form the imino propyl zirconium species **B** (ii). Subsequently, intermediate **B** reacts with HBpin via C−Zr and H−B bond metathesis to give **C** and regenerate Zr−H species (iii). In the second catalytic cycle, Zr−H hydride transfer to intermediate **C** gives intermediate **D** (iv), which is further reduced by the previously released H₂ to **2a** with hydrogenolysis or H₂ metathesis (v). The last step is supported by the experiment that when we introduced 1 atm of deuterium gas into the standard reaction, 29% deuterium labeling at the α-carbon adjacent to N−H of **2a** could be

obtained (Fig. 5G, Supplementary Fig. 10), suggesting that hydrogen metathesis occurred[99–101].

In summary, an unprecedented zirconium- and hafnium-catalyzed C−C single bond activation and subsequent hydroboration is realized using a catalytic system based on Cp₂ZrCl₂ and Cp₂HfCl₂. Our catalytic approach applies to various cyclopropylamines. Selective cleavage of the proximal C$_α$−C$_β$ single bond was achieved, with the tolerance of multiple functional groups as well as bio- and medicine-derived substrates. Mechanistic studies disclose that the in-situ generated Zr−H

**Fig. 6 | Proposed reaction pathway for the Zr-catalyzed C−C single bond hydroboration.** Key steps for the transformation: i) N−H bond metathesis; ii) C−C activation; iii) B−H bond metathesis, iv) Zr−H hydride transfer; v) H₂ metathesis.

species and the free N−H group of the substrates play key roles in this transformation via Zr−H and N−H metathesis to form N−Zr$^{IV}$ species, and the subsequent C−C single bond activation is realized via a β-carbon elimination route. Our work presents an unprecedented group IV metal-catalyzed C−C single bond activation and hydroboration reaction. The C−C single bond activation model that was well studied for late-transition metals, were also elaborated to be applicable to the group IV metals.

## Methods

### General procedure for the Zr-catalyzed hydroboration of cyclopropylamines

In a nitrogen-filled glovebox, to a 15 mL pressure tube with a magnetic stirrer was added catalytic amount of Cp₂ZrCl₂ (0.01 mmol, 2.9 mg), K₂CO₃ (0.06 mmol, 8.3 mg), corresponding cyclopropylamine substrates (0.2 mmol), HBpin (0.3 mmol, 43.5 μL), and toluene (1 mL) in a sequence manner. Then, the pressure tube was taken out of the glove box and allowed to stir at 120 °C for 24 h. Upon completion, all the solvent was evaporated, and the crude product was isolated on silica gel using flash chromatography with dichloromethane/ethyl acetate as the eluent to give the corresponding products.

## Data availability

Experimental details, Synthetic Procedures, Tables for condition optimizations (Supplementary Tables 1–5), Figures for mechanistic studies, and DFT calculations, NMR spectra (Supplementary Figs. 1–248), products characterizations are included in the Supplementary Information. All other data are available from the corresponding author upon request.

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

## Acknowledgements

We are grateful to the National Natural Science Foundation of China (22271295, L.W.); Gansu Provincial Natural Science Foundation Key Project (23JRRA606, L.W.), Major Program of the Lanzhou Institute of Chemical Physics, CAS (No. ZYFZFX-9, L.W.); State Key Laboratory Program of the Lanzhou Institute of Chemical Physics (CHGZ-202208, L.W.) for generous financial support.

## Author contributions

L.Wu, S. Li conceived the project and designed the experiments. S. Li performed the experiments and analyzed the data. H. Jiao performed computational chemistry. L.Wu, H. Jiao and X. Shu wrote the manuscript. All the authors discussed the results and commented on the manuscript.

## Competing interests

The authors declare no competing interests.
