## [Peer Review File · Nature Communications]

Zirconium and Hafnium Catalyzed C–C Single Bond HydroborationReviewers' Comments:

Reviewer #1:

Remarks to the Author:

Wu and coworkers presented a technique for Zr-catalyzed hydroboration of cyclopropylamines which resulted in the formation of γ -boronated amides in moderate to good yields. The methodology was found to be compatible with a range of functional groups. However, the absence of demonstrations involving other polar groups such as alkynyl, amido, amino, hydroxyl, etc., in various biologically active molecules diminishes the significance of this research. Additionally, the inclusion of sterically hindered secondary amides, featuring a quaternary carbon substituent adjacent to the carbonyl group, appears to be crucial to prevent potential amide reduction as an unwanted side reaction. Nowadays, it is more demanding to cleave the C-C bond of less strained cyclic rings (4, 5, or 6-membered). In addition to questioning the novelty of this work, all characterizations of cyclopropylamines as used starting materials are missed in Supporting Information. Moreover, the peak intensities observed in ^1H - and ^{13}C -NMR spectra are too low in the report.

Regarding the mechanism of C-C bond activation, the authors propose two possible pathways. One involves the formation of an amido zirconium (IV) intermediate, followed by β -carbon elimination. The other one suggests oxidative insertions of zirconium (II) into cyclopropane. However, the existing studies on the mechanism are far from comprehensive and informative. It is important to note that Cantat's work, published in *Angewandte Chemie* (2022, 61, e202206170), supports the observation of N-Zr species with the release of H_2 and should be appropriately referenced in the discussion. The assignment of Cp_2ZrCO_3 solely based on ^1H -NMR shown fig. 4Cb is inaccurate. Additionally, the lack of isolated intermediates B, C, and D undermines the rationalization of the proposed catalytic cycle (fig. 5). Demonstration of carbozirconocene in Scheme 2b through HRMS is insufficient to confirm the claimed structure. Considering the aforementioned comments, I don't think that this manuscript is suitable for publication in *Nature Communications*.

Reviewer #2:

Remarks to the Author:

In this manuscript, Wu, Jiao and co-workers developed a catalytic hydroboration of C-C single bond of cyclopropylamines with group 4 metallocene complexes. This reaction provides a facile access to a variety of γ -boronated amines, which are otherwise difficult to obtain. It has impressed me most for the use of earth-abundant and cheap early transition metal-based catalyst for such activation and transformation. Moreover, the authors also performed a number of control experiments to elucidate the possible reaction mechanism. Considering the methodological novelty and high quality, I recommend publication of this manuscript in *Nature Communication* after addressing the following issues:

1. The authors should test Cp_2ZrH_2 as a catalyst in the condition optimization (Table 1).
2. Is this hydroboration reaction also applicable for the cyclobutyl-substituted amide?
3. The authors should conduct the catalytic reaction of 1a with HBpin under an atmosphere of H_2 or D_2 . This may further support that metathesis reaction does occur in the catalytic cycles.

Other minor issues

4. In table 1 and scheme 2, catalyst should be used with 5 mol%, NOT 5 mmol%.
5. Page 2, the sentence "Compared with late-transition metal complexes, early transition metal complexes often have different structures" made no sense.
6. In figure 1, the structure of compound 2y looks weird.
7. Figure 4 is too small to clearly see the reactions and NMR spectra.
8. In SI, all ^{13}C NMR should be " $^{13}\text{C}\{^1\text{H}\}$ NMR".

Reviewer #3:

Remarks to the Author:

Wu, Jiao, and coworkers report a Zr-catalyzed, HBin-mediated ring-opening of N-cyclopropylamides through C-C cleavage of the cyclopropyl ring to deliver gamma-amino alkylboronate derivatives. While largely limited to N-pivalyl derivatives, the scope is otherwise wide, and the reaction is efficient. The authors also report that the transformation not only works for zirconocene dichloride based catalyst system but also for hafnocene dichloride as well. Though the rationale isn't clear, the authors report certain synthetically useful differences in scope when Cp₂HfCl₂ is used (e.g., better tolerance of nitrile-containing substrates). This represents a relatively rare report of hafnium used in catalysis.

While the synthetic results are suitable for publication in Nature Communications, a substantial revision of the introduction and mechanistic conclusion is needed before it can be further considered for publication. For publication, the authors should address each of the points below:

- 1) "Besides these two precedents, no other reports for C-C single bond activation using group IV metals are known." This is just not true. Even a cursory search for examples by a non-expert like myself revealed the following early and recent reports: Dimmock, J Chem Soc Chem Commun 1994, 2323; Hull Organometallics 2015, 34, 4190; Tonks, Organometallics 2023, doi:acs.organomet.3c00032. I strongly suspect that many other examples can be found with a more thorough search.
- 2) Negishi's reagent and Rosenthal's reagent need to be defined.
- 3) "Then, in 2014, Marek reported an expedient catalytic approach, including allylic C-H activations followed by C-C single bond activation." This is also not a true statement. This paper is stoichiometric in Zr.
- 4) In Figure 5, it is claimed that in step (v), H₂ is used to regenerate Cp₂ZrHX. Is there literature precedent or experimental results to support this sigma-bond metathesis with H₂? Otherwise, it makes much more sense that HBpin is the reagent that does this. The N-Bpin bond would be hydrolyzed to give the N-H upon workup.
- 5) The evidence for the direct C-C insertion mechanism is weak at best. In particular, the zirconacyclobutane has the same molecular weight as the corresponding gamma, delta-imino zirconocene hydride, so "detection by HRMS" means little. One can imagine the (highly basic) Zr-C bonds of the Rosenthal reagent undergoing protonolysis by substrate 1a, leading to Cp₂Zr[N]₂ as the intermediate, where [N] is an amido ligand derived from 1a. This could then under a mechanism similar in nature to the one shown in Figure 5. If this mechanism is correct, the N-methylated substrate 1a-Me should also work (or at least be able to form the zirconacyclobutane intermediate). Overall, while this mechanism is not excluded by the current data, there needs to be stronger evidence before it can be claimed as a likely alternative activation pathway.

Responses Letter

Reviewer 1 (R1):

Recommendation: Wu and coworkers presented a technique for Zr-catalyzed hydroboration of cyclopropylamines which resulted in the formation of γ -boronated amides in moderate to good yields. The methodology was found to be compatible with a range of functional groups. However, the absence of demonstrations involving other polar groups such as alkynyl, amido, amino, hydroxyl, *etc.*, in various biologically active molecules diminishes the significance of this research. Additionally, the inclusion of sterically hindered secondary amides, featuring a quaternary carbon substituent adjacent to the carbonyl group, appears to be crucial to prevent potential amide reduction as an unwanted side reaction. Nowadays, it is more demanding to cleave the C-C bond of less strained cyclic rings (4, 5, or 6-membered). In addition to questioning the novelty of this work, all characterizations of cyclopropylamines as used starting materials are missed in Supporting Information. Moreover, the peak intensities observed in ^1H - and ^{13}C -NMR spectra are too low in the report.

Regarding the mechanism of C-C bond activation, the authors propose two possible pathways. One involves the formation of an amido zirconium (IV) intermediate, followed by β -carbon elimination. The other one suggests oxidative insertions of zirconium (II) into cyclopropane. However, the existing studies on the mechanism are far from comprehensive and informative. It is important to note that Cantat's work, published in *Angewandte Chemie* (2022, 61, e202206170), supports the observation of N-Zr species with the release of H_2 and should be appropriately referenced in the discussion. The assignment of Cp_2ZrCO_3 solely based on ^1H -NMR shown fig. 4Cb is inaccurate. Additionally, the lack of isolated intermediates B, C, and D undermines the rationalization of the proposed catalytic cycle (fig. 5). Demonstration of carbozirconocene in Scheme 2b through HRMS is insufficient to confirm the claimed structure. Considering the aforementioned comments, I don't think that this manuscript is suitable for publication in *Nature Communications*.

Our Response: We thank R1 very much for all the constructive comments and suggestions that guide us in improving the quality of our work. Consequently, in this revised manuscript, we have made every effort to resolve all issues raised by R1 (please see our revision for details). We hope that our revision meets the standard for publication in *Nat. Commun.*

Detailed Comments and Response:

1. The absence of demonstrations involving other polar groups such as alkynyl, amido, amino, hydroxyl, *etc.*, in various biologically active molecules diminishes the significance of this research.

Our Response: We thank R1 for this comment. It is noted that pinacolborane (HBpin) used as the boration reagent in our system, can react with alkynyl, amido, amino, hydroxyl groups with or

without a catalyst. Therefore, these polar functional groups are barely tolerated in most the reported hydroboration systems that use HBpin.

Nevertheless, we have synthesized several substrates containing various functional groups as suggested and tried them in our system. We found that alkynyl, thioether, boronate, silyl, sulfonamide, and nitrile groups are tolerated (Scheme R1). We have added those substrates in the revised manuscript in Figure 1 and Figure 2.

Newly added substrates with various functional groups

Unsuccessful substrates

Scheme R1. Substrates scope studies with various functional groups.

2. The inclusion of sterically hindered secondary amides, featuring a quaternary carbon substituent adjacent to the carbonyl group, appears to be crucial to prevent potential amide reduction as an unwanted side reaction.

Our Response: We thank R1 for this comment. We have tried cyclohexyl- and phenyl-substituted substrates in our systems, and the corresponding hydroboration products (**2aq**, **2ar**) were obtained without problems (Scheme R2). Those data were included in the revised Figure 2.

Scheme R2. Non-quaternary carbon substituted substrates.

3. It is more demanding to cleave the C-C bond of less strained cyclic rings (4,5, or 6-membered).

Our Response: We thank R1 for this comment. The cleavage of less strained cyclic rings (4, 5, or 6-membered) is a demanding but largely undissolved area. Nowadays, only the cleavage of C-C bond of biphenylenes, cyclobutenediones, benzocyclobutenediones, or cyclobutanones is studied. Thus, the cleavage of unfunctionalized 4-6 cyclic rings (alkanes) using homogeneous transition

metal catalysis is still challenging. Nevertheless, we synthesized the 4, 5, or 6-membered ring systems and tried them in our system; all gave traces or no expected reactions (Scheme R3).

Scheme R3. 4,5, or 6-membered cyclic rings tested in our system.

4. All characterizations of cyclopropylamines as used starting materials are missed in Supporting Information. Moreover, the peak intensities observed in ^1H - and ^{13}C -NMR spectra are too low in the report.

Our Response: We thank R1 for pointing this out. We have included all the characterizations of cyclopropylamines starting substrates in the revised Supporting Information. The ^1H - and ^{13}C -NMR spectra were remeasured to show the peaks more clearly in Supporting Information. Please see the revised Supporting Information for a more detailed revision.

5. Regarding the mechanism of C-C bond activation, the authors propose two possible pathways. One involves the formation of an amido zirconium (IV) intermediate, followed by β -carbon elimination. The other one suggests oxidative insertions of zirconium (II) into cyclopropane. However, the existing studies on the mechanism are far from comprehensive and informative. It is important to note that Cantat's work, published in *Angewandte Chemie* (2022, 61, e202206170), supports the observation of N-Zr species with the release of H_2 and should be appropriately referenced in the discussion. The assignment of Cp_2ZrCO_3 solely based on ^1H -NMR shown fig. 4Cb is inaccurate. Additionally, the lack of isolated intermediates B, C, and D undermines the rationalization of the proposed catalytic cycle (fig. 5). Demonstration of carbozirconocene in Scheme 2b through HRMS is insufficient to confirm the claimed structure.

Our Response: We thank R1 for all the comments, which helped us significantly improve the quality of our work.

We are sorry that we did not make this clear in our previous version that the two reaction pathways were proposed for two different systems: 1) $\text{Zr}^{\text{IV}}\text{-H}$ species as the active catalyst, which activates C-C bond via the β -carbon elimination route (Figure R1); and 2) Zr^{II} species as the active catalyst which activate C-C bond via direct oxidative addition route. The latter was just some additional findings using Zr^{II} species [Rosenthal's reagent, $\text{Cp}_2\text{Zr}(\text{PMe}_3)_2$] that have nothing to do with our main catalytic system ($\text{Cp}_2\text{ZrCl}_2/\text{K}_2\text{CO}_3$) discussed in this manuscript. Thus, to avoid this misunderstanding, we deleted the "Cp₂Zr species-catalyzed reaction" section from our manuscript to make the mechanism clear for readers to follow.

Figure R1. Proposed reaction pathway: i) N–H bond metathesis; ii) C–C activation; iii) B–H bond metathesis, iv) Zr–H hydride transfer; v) H₂ metathesis.

In addition, we have provided more information to support our Zr^{IV}–H species-mediated β -carbon elimination route. First, the suggested reference from Cantat's work and references cited therein were cited and discussed in our manuscript, supporting the formation of species **A** in our proposed catalytic cycle. We have also conducted additional experiments by reacting Cp₂ZrHCl with **1a** and found the release of H₂ or HD when **1a-D** was used (Figure R2).

Figure R2. ¹H NMR spectra show the release of H₂ or HD by reacting Cp₂ZrHCl with **1a**.

Then, to further prove the importance of the N–H functional group, *N*-methylated analog **1a-Me**, and substrates by replacing the N–H group with CH₂ and O groups (**1a-C** and **1a-O**) were subjected to our reaction conditions. No corresponding C–C bond hydroboration product was observed as expected (Scheme R4).

Schem R4. Control experiments to show the importance of N–H bond.

For the assignment of Cp_2ZrCO_3 , we have isolated and characterized it using IR spectroscopy. In the IR spectrum, peaks at 1597 and 1350 cm^{-1} could be assigned to the asymmetric stretching mode of CO_3^{2-} ; peaks at 1097 and 864 cm^{-1} are designated to the symmetric stretching mode of CO_3^{2-} . The band at 660 cm^{-1} is attributed to the stretching vibration of Zr–O (Figure R3). Those data are in accordance with the literature reports (*ChemSusChem*. **2015**, *8*, 4022–4029; *Ind. Eng. Chem. Res.* **2018**, *57*, 10126–10136).

Figure R3. IR spectrum of “ Cp_2ZrCO_3 ”.

In addition to the above-mentioned experiments and characterizations. We also attempted to isolate intermediates **B**, **C**, and **D**. Fortunately and to our delight, we observed **2u'** and **2ap'**, intermediate **C**-type compounds on GC/MS in the reaction of substrates **1u** and **1ap** (Figure R4). The structure of **2u'** was further confirmed by observing the imine carbon signal peak on the crude ^{13}C NMR spectrum (Figure R5). These two experiments support the existence of intermediate **B** and **C**.

Figure R4. GC/MS chromatogram of **2u'** and **2ap'**.

Figure R5. Detection of imine intermediate **2u'** by crude ^{13}C NMR spectroscopy.

In addition to approving the existence of intermediate **B** and **C**, we further conducted the deuterium labeling experiments by introducing 1 atm of deuterium gas into the standard reaction to support the presence of intermediate **D**. The ^1H NMR spectrum of the obtained products **2a-D** indicates a 29% deuterium-labeling at the α -carbon adjacent to N–H of **2a** (Figure R6). This phenomenon suggests the existence of H_2 metathesis process in the reaction with the presence of intermediate **D**.

Moreover, we also computed the β -carbon elimination route for the C–C bond activation (Figure R7). Firstly, we calculated H_2 release from Zr–H and N–H metathesis, which has a Gibbs free energy barrier of 5.79 kcal/mol (**TS1**) and is exergonic by 14.00 kcal/mol (**A**). Next, activation of the C–C bond via β -carbon elimination has a Gibbs free energy barrier of 27.18 kcal/mol (**TS2**) and is endergonic by 21.49 kcal/mol (**B**). Overall, the C–C bond cleavage is endergonic by 7.49 kcal/mol and has an apparent Gibbs free energy barrier of 27.18 kcal/mol, which is compatible with our reaction conditions. We have included the calculation results in the revised Supporting Information.

Overall, we have conducted additional control experiments, substrate synthesis, IR and NMR studies, and DFT calculations to address R1's concerns about our reaction mechanism and other comments. A precise reaction mechanism is given with further evidence for the existence of intermediates **A**, **B**, **C** and **D**.

Therefore, we hope that our revision is sufficient and suitable. We hope we can persuade R1 to agree with us on the importance of our work.

Figure R6. ^1H NMR spectrum of **2a-D** obtained by deuterium labeling experiment with D_2 in the standard reaction.

Figure R7. BP86-D3-SCRF/TZVP Gibbs free energy diagram of the $\text{Zr}^{\text{IV}}\text{-H}$ -catalyzed C-C bond cleavage (ΔG , kcal/mol).

Reviewer 2 (R2):

Recommendation: In this manuscript, Wu, Jiao and co-workers developed a catalytic hydroboration of C-C single bond of cyclopropylamines with group 4 metallocene complexes. This reaction provides a facile access to a variety of γ -boronated amines, which are otherwise difficult to obtain. It has impressed me most for the use of earth-abundant and cheap early transition metal-based catalyst for such activation and transformation. Moreover, the authors also performed a number of control experiments to elucidate the possible reaction mechanism. Considering the methodological novelty and high quality, I recommend publication of this manuscript in *Nature Communication* after addressing the following issues.

Our Response: We thank R2 for the positive comments and helpful suggestions to improve the quality of our manuscript. We have carefully revised our manuscript based on R2's suggestions. Please see the following detailed points-to-points responses.

Comments and Response:

1. The authors should test Cp_2ZrH_2 as a catalyst in the condition optimization (Table 1).

Our Response: We thank R2 for this suggestion. We tested 5 mol% Cp_2ZrH_2 as catalyst, and product **2a** was obtained in 95% GC yield. This data was included in the revised Table 1 as entry 10 with the following description "In addition, 95% yield of **2a** was obtained using 5 mol% Cp_2ZrH_2 as catalyst without K_2CO_3 (Table 1, entry 10). However, considering the simplicity of using readily available and inexpensive Cp_2ZrCl_2 as the catalyst, we conducted the following studies using $\text{Cp}_2\text{ZrCl}_2/\text{K}_2\text{CO}_3$ system (Table 1, entry 9). The results with Cp_2ZrH_2 gave us some clues for the subsequent mechanism studies (*vide infra*)".

2. Is this hydroboration reaction also applicable for the cyclobutyl-substituted amide?

Our Response: We have tried the cyclobutyl-substituted amide in our reaction. However, no corresponding hydroboration product was obtained.

3. The authors should conduct the catalytic reaction of **1a** with HBpin under an atmosphere of H_2 or D_2 . This may further support that metathesis reaction does occur in the catalytic cycles.

Our Response: We thank R2 for this helpful suggestion. We have conducted the deuterium labeling experiments by introducing 1 atm of deuterium gas into the standard reaction. From the ^1H NMR spectrum of the obtained products **2a-D**, we can see 29% deuterium-labeling at the α -carbon adjacent to N-H of **2a** (Figure R8). This phenomenon further supports the existence of H_2 metathesis process in the reaction.

Figure R8. ^1H NMR spectrum of **2a-D** obtained by deuterium labeling experiment with D_2 .

4. In table 1 and scheme 2, catalyst should be used with 5 mol%, NOT 5 mmol%.

Our Response: We thank R2 for pointing this out. We have corrected those errors.

5. Page 2, the sentence “Compared with late-transition metal complexes, early transition metal complexes often have different structures” made no sense.

Our Response: We thank R2 for pointing this out. We have revised this sentence to “Early-transition metals have different electron configurations with late ones. Thus, their complexes often show different or orthogonal reactivities with late-transition metal complexes”.

6. In figure 1, the structure of compound **2y** looks weird.

Our Response: We thank R2 for pointing this out. Compound **2y** is now compound **2ac** in our revised manuscript, the structure of which is redrawn.

7. Figure 4 is too small to clearly see the reactions and NMR spectra.

Our Response: We thank R2 for pointing this out. We have enlarged Figure 4 in the revised manuscript.

8. In SI, all ^{13}C NMR should be “ $^{13}\text{C}\{^1\text{H}\}$ NMR”.

Our Response: We thank R2 for pointing this out. We have replaced all the ^{13}C NMR to $^{13}\text{C}\{^1\text{H}\}$ NMR in the SI.

Reviewer 3 (R3):

Recommendation: Wu, Jiao, and coworkers report a Zr-catalyzed, HBpin-mediated ring-opening of N-cyclopropylamides through C-C cleavage of the cyclopropyl ring to deliver gamma-amino alkylboronate derivatives. While largely limited to *N*-pivalyl derivatives, the scope is otherwise wide, and the reaction is efficient. The authors also report that the transformation not only works for zirconocene dichloride based catalyst system but also for hafnocene dichloride as well. Though the rationale isn't clear, the authors report certain synthetically useful differences in scope when Cp₂HfCl₂ is used (e.g., better tolerance of nitrile-containing substrates). This represents a relatively rare report of hafnium used in catalysis. While the synthetic results are suitable for publication in *Nature Communications*, a substantial revision of the introduction and mechanistic conclusion is needed before it can be further considered for publication.

Our Response: We thank R3 for agreeing with us on the importance of our work. We also appreciate all the helpful suggestions to improve the quality of our work. We have carefully revised our manuscript based on R3's suggestions. Please see the following detailed points-to-points responses.

Comments and Response:

1. "Besides these two precedents, no other reports for C-C single bond activation using group IV metals are known." This is just not true. Even a cursory search for examples by a non-expert like myself revealed the following early and recent reports: Dimmock, *J. Chem. Soc. Chem. Commun.*, 1994, 2323; Hull, *Organometallics*, 2015, 34, 4190; Tonks, *Organometallics*, 2023, doi: acs.organomet.3c00032. I strongly suspect that many other examples can be found with a more thorough search.

Our Response: We thank R3 for pointing this out and the suggested references. The last two references suggested by R3 were using titanium complexes. The paragraph before this description talked about zirconium, so we meant to say, "Besides these two precedents, no other reports for C-C single bond activation using zirconium are known." Nevertheless, we thoroughly searched using zirconium for C-C single bond activation, and we found more examples and cited them in our revised manuscript. However, all the examples (including titanium) used (over) stoichiometric amounts of zirconium without any catalytic examples. This is in sharp contrast with our catalytic work, highlighting the importance of our method. The corresponding sentence was revised to "C-C single bond cleavage by a zirconium species has also been reported intermittently since the 1990s.⁷⁰⁻⁷³ In 1994, Rosenthal described the activation of conjugated C-C single bonds of a 1,3-butadiene moiety (C≡C-C≡C) using Rosenthal's reagent, resulting in a dimeric complex (Scheme 1b).⁷⁴ Dimmock and Whitby also found that zirconocene η²-alkene and η²-imine complexes with adjacent cyclopropane rings could undergo cyclopropane ring cleavage.⁷⁵ Then, in 2014 Marek reported an expedient approach, including allylic C-H activations followed by C-C single bond activation (Scheme 1c).⁷⁶ It is noted that all the former instances used (over) stoichiometric

amounts of zirconium, and no precedents of catalytic methods had been developed—to the best of our knowledge. Consequently, activating C–C single bonds with zirconium catalysis for chemical transformation remains a significant challenge. It is of considerable scientific and practical interest to synthetic organic chemistry to address this”.

2. Negishi’s reagent and Rosenthal’s reagent need to be defined.

Our Response: We thank R3 for this suggestion. We have defined Negishi’s reagent and Rosenthal’s reagent in the main text as “such as Negishi’s (“Cp₂ZrBu₂”)⁶² and Rosenthal’s reagents (“Cp₂Zr(py)Me₃SiC≡CSiMe₃”),⁶³” and cited two literatures for their preparation (*Tetrahedron Letters*, **1984**, 27, 2829-2832; *Angew. Chem. Int. Ed.* **1993**, 32, 1193-1195.).

3. “Then, in 2014, Marek reported an expedient catalytic approach, including allylic C-H activations followed by C-C single bond activation.” This is also not a true statement. This paper is stoichiometric in Zr.

Our Response: We thank R3 for pointing this out. This statement has been corrected. All the previous literature on using zirconium for C–C single bond activation is in (over) stoichiometric amount, including this work from Marek in 2014 (*Nature* **505**, 199-203), which further highlighted the importance of our work.

4. In Figure 5, it is claimed that in step (v), H₂ is used to regenerate Cp₂ZrHX. Is there literature precedent or experimental results to support this sigma-bond metathesis with H₂? Otherwise, it makes much more sense that HBpin is the reagent that does this. The N-Bpin bond would be hydrolyzed to give the N-H upon workup.

Our Response: We have both literature and experiment support for this step. The following three papers describe the sigma-bond metathesis of alkyl zirconium complexes with H₂ (*J. Am. Chem. Soc.* **1978**, 100, 3246-3248; *J. Am. Chem. Soc.* **1982**, 104, 1846-1855; *Organometallics*, **1987**, 6, 1041-1051). In addition, we have conducted the deuterium labeling experiments by introducing 1 atm of deuterium gas into the standard reaction. From the ¹H NMR spectrum of the obtained product **2a-D**, we can see 29% deuterium-labeling at the α-carbon adjacent to N–H of **2a** (Figure R9). This phenomenon supports the existence of H₂ metathesis process in the reaction. Moreover, we used only 1.5 equivalent of HBpin for the transformation. If the HBpin plays the role of catalyst regeneration, we should need at least 2 equivalents of HBpin.

Figure R9. ^1H NMR spectrum of **2a-D** obtained by deuterium labeling experiment with D_2 .

5. The evidence for the direct C-C insertion mechanism is weak at best. In particular, the zirconacyclobutane has the same molecular weight as the corresponding gamma, delta-imino zirconocene hydride, so "detection by HRMS" means little. One can imagine the (highly basic) Zr-C bonds of the Rosenthal reagent undergoing protonolysis by substrate **1a**, leading to $\text{Cp}_2\text{Zr}[\text{N}]_2$ as the intermediate, where [N] is an amido ligand derived from **1a**. This could then under a mechanism similar in nature to the one shown in Figure 5. If this mechanism is correct, the *N*-methylated substrate **1a-Me** should also work (or at least be able to form the zirconacyclobutane intermediate). Overall, while this mechanism is not excluded by the current data, there needs to be stronger evidence before it can be claimed as a likely alternative activation pathway.

Our Response: We thank R3 for pointing this out. We are sorry that we did not make this clear in our previous version that the two reaction pathways were proposed for two different systems, not two alternative activation pathways: 1) when using $\text{Zr}^{\text{IV}}\text{-H}$ species as the active catalyst, the activate C-C bond was via β -carbon elimination route by forming N-Zr species. This is the key finding of our work; and 2) when using Zr^{II} species as the active catalyst, the activation of C-C bond was via direct oxidative addition. The latter was some additional findings using Zr^{II} species [Rosenthal's reagent, $\text{Cp}_2\text{Zr}(\text{PMe}_3)_2$], which has nothing to do with our reported catalytic system ($\text{Cp}_2\text{ZrCl}_2/\text{K}_2\text{CO}_3$). Thus, to avoid this misunderstanding, we removed the "Cp₂Zr species-catalyzed reaction" section from our manuscript. Furthermore, we have conducted additional control experiments, substrate synthesis, IR, and NMR studies to support our reaction mechanism further

(Figure 5 in the main text). A precise reaction mechanism with more support is given in the revised manuscript. We have copied our mechanistic study section herein for R3's reference:

Mechanistic studies in the manuscript:

To shed light on the reaction mechanism, several control experiments were performed (Figure 4). The possible formation of an alkene intermediate via ring-opening of cyclopropanes followed by hydroboration was studied. However, no alkenes were detected after 3 or 12 h under standard reaction conditions with or without HBpin (Figure 4Aa, Figure S1). Utilization of alkenes **1a'** and **1a''** afforded less than 6% **2a** (Figure 4Ab). When enantioenriched substrate (1*S*, 2*R*)-**1as** was applied, the desired product (*R*)-**2as** was obtained without erosion of the enantioselectivities (Figure 4Ac, Figures S2 and S3). Those results excluded a consecutive cyclopropane ring opening-hydroboration process. Then, the possibility of a reaction pathway that involved a radical species was investigated. TEMPO (2,2,6,6-tetramethylpiperidinyloxy) (1–2 equiv) had almost no effect on the results. However, upon increasing the amount thereof (4 equiv.) the yields of **2a** decreased to 44% (Figure 4B). At this point, it should be borne in mind that TEMPO inhibition experiments can sometimes provide ambiguous results.⁸⁵ Thus, additional experiments with the addition of 9,10-dihydroanthracene (DHA) were conducted; no effect on the yield of **2a** was detected (Figure 4B). The results with TEMPO and DHA excluded a radical mechanism.

Then, the active zirconium catalytic species was studied. Upon the combination of Cp₂ZrCl₂ and K₂CO₃ in d⁸-Tol heated at 120 °C for 12 h, a new species appeared around 6.0 ppm in the ¹H NMR spectrum (Figure 4Ca). With 2 equiv. K₂CO₃ and heating for a longer reaction time, the Cp₂ZrCl₂ was fully converted to this new species (Figure 4Cb). Then, the isolated new species was characterized by IR spectroscopy and was currently assigned to "Cp₂ZrCO₃" by comparison with literature data (Figure S4).^{86,87} Nevertheless, upon further adding HBpin to the above solution, we could detect the formation of Zr–H species in the ¹H NMR spectrum by trapping with acetone (Figure 4Cc, Figure S5). This finding, together with the fact that Cp₂ZrHCl or Cp₂ZrH₂ can catalyze the C–C bond hydroboration process without K₂CO₃ (65% and 95% yields, Table S2), we concluded that Zr–H species are essentially the active catalysts via the consecutive reactions of Cp₂ZrCl₂, K₂CO₃, and HBpin (Figure 4Cd). According to the work from Ganem,⁸⁸ Rosenthal,⁸⁹ and Cantat,⁹⁰ the active Zr–H species can interact with **1a** to form N–Zr species via metathesis with N–H bond. This is further proved in our case that Cp₂ZrHCl reacts with the N–H group of **1a** with the release of H₂ or HD when **1a-D** was used (Figure 4D, S6). To add further proof of the importance of the N–H, *N*-methylated analogue substrate **1a-Me**, and replace the N–H with CH₂ or O substrates **1a-C**, **1a-O** were subjected to our reaction conditions. As expected, no corresponding C–C bond hydroboration product were observed (Figure 4E). Keep in mind that β-carbon elimination is one of the main pathway for C–C bond cleavage. It is natural to think that after the formation of N–Zr species, a β-carbon elimination may proceed to cleavage the C–C bond to produce an imino propyl zirconium species. This is consistent with the fact that we can observe the

presence of the putative imine intermediate both on GC/MS and ^{13}C NMR when substrate **1u** was used (Figure 4F, S7-8).

Figure 4. Mechanism studies: A) possible alkene intermediate formation; B) possible radical pathway; C) ^1H NMR spectra show the formation of Zr-H species; D) ^1H NMR spectra show the release of H_2 or HD by reacting Cp_2ZrHCl with **1a**; E) Control experiments to show the importance of N-H bond; F) Detection of imine intermediate **2u'** by crude ^{13}C NMR spectroscopy; G) Deuterium labeling experiment by introducing D_2 in the standard reaction.

Based on the above mechanistic study and our DFT calculation results (Figure S9), we conclude the following general reaction pathway for our $\text{Cp}_2\text{ZrCl}_2/\text{K}_2\text{CO}_3$ system (Figure 5). First, the in-situ formed Zr–H species reacts with N–H bonds of the substrates to form N–Zr^{IV} species **A** via H_2 release (i). Next, the C–C single bond is cleaved via β -carbon elimination of intermediate **A** to form the imino propyl zirconium species **B** (ii). Subsequently, intermediate **B** reacts with HBpin via C–Zr and H–B bond metathesis to give **C** and regenerate Zr–H species (iii). In the second catalytic cycle, Zr–H hydride transfer to intermediate **C** gives intermediate **D** (iv), which is further reduced by the previously released H_2 to **2a** with hydrogenolysis or H_2 metathesis (v). The last step is supported by the experiment that when we introduced 1 atm of deuterium gas into the standard reaction, 29% deuterium labeling at the α -carbon adjacent to N–H of **2a** could be obtained (Figure 4G, S10), suggesting that hydrogen metathesis occurred.⁹¹⁻⁹³

Figure 5. Proposed reaction pathway: i) N–H bond metathesis; ii) C–C activation; iii) B–H bond metathesis, iv) Zr–H hydride transfer; v) H_2 metathesis.

Moreover, we also carried out computational studies for the β -carbon elimination route for the C–C bond activation (Figure R10). Firstly, we computed H_2 release from Zr–H and N–H metathesis, which has a Gibbs free energy barrier of 5.79 kcal/mol (TS1) and is exergonic by 14.00 kcal/mol (A). Next, activation of the C–C bond via β -carbon elimination has a Gibbs free energy barrier of 27.18 kcal/mol (TS2) and is endergonic by 21.49 kcal/mol (B). Overall, the C–C bond cleavage is endergonic by 7.49 kcal/mol and has an apparent Gibbs free energy barrier of 27.18 kcal/mol, which is compatible with our reaction conditions. We have included the calculation results in the revised Supporting Information.

Figure R10. BP86-D3-SCRF/TZVP Gibbs free energy diagram of the $\text{Zr}^{\text{IV}}\text{H}$ -catalyzed C-C bond cleavage (ΔG , kcal/mol).

Overall, we have conducted additional control experiments, substrate synthesis, IR and NMR studies, and DFT calculations to address R3's concerns about our reaction mechanism and other comments. A precise reaction mechanism with more support is given in the revised manuscript. We hope that our revision is sufficient and suitable. We hope we can persuade R3 to agree with us on the importance of our work.

Reviewers' Comments:

Reviewer #1:

Remarks to the Author:

In this revision, Wu and coworkers have significant improvements on the Zr-H and Hf-H catalyzed hydroboration of cyclopropylamine derivatives. They have devoted more effort to demonstrating functional group compatibility, making it easier and more intuitive for readers to grasp the limitations of this approach. Eventually applications are still limited to the ring-opening of the cyclopropyl motif, even when catalysis were conducted at high temperatures. Considering this zirconium catalysis as a first example enabling C-C bond hydroboration, it is interesting in the field. The mechanistic insights are now much clearer than in the original submission, and the supporting data adequately meet the publication standards. Therefore, I believe this work is now more appropriate in consideration for publication in Nature Communications.

Note: please check figure 4Cd, the corresponding zirconium product was missing in equation.

END

Reviewer #2:

Remarks to the Author:

Reviewer thanks the authors for addressing all comments, in some cases even doing additional experiments. In the present text this has led to satisfactory extra information. This excellent paper can be advised for publication without modifications.

Reviewer #3:

Remarks to the Author:

The mechanistic issues I previously raised have been adequately addressed by the authors, but I still have some issues with the way the introduction is written.

For additional examples of stoichiometric Zr-mediated C-C bond cleavage, Bull. Chem. Soc. Jpn. 1999, 72, 2591 by Takahashi ("Carbon-Carbon Bond Cleavage and Selective Transformations of Zirconacycles") and references therein should be cited, and the strategy of reductive cleavage of zirconacycles should be mentioned in the main text.

For catalytic alkane C-C bond cleavage and alkane metathesis by immobilized Zr catalysts, the work of Basset and coworkers should be mentioned: Acc. Chem. Res. 2010, 43, 323 and Angew. Chem. Int. Ed. 1998, 37, 806.

Other than this, the zirconocene hydride product for Figure 4Cd is missing.

Some copy editing for idiomatic usage of English should be done, but otherwise, the manuscript is suitable for publication.

Responses Letter

Reviewer 1 (R1):

Recommendation: In this revision, Wu and coworkers have significant improvements on the Zr-H and Hf-H catalyzed hydroboration of cyclopropylamine derivatives. They have devoted more effort to demonstrating functional group compatibility, making it easier and more intuitive for readers to grasp the limitations of this approach. Eventually applications are still limited to the ring-opening of the cyclopropyl motif, even when catalysis were conducted at high temperatures. Considering this zirconium catalysis as a first example enabling C-C bond hydroboration, it is interesting in the field. The mechanistic insights are now much clearer than in the original submission, and the supporting data adequately meet the publication standards. Therefore, I believe this work is now more appropriate in consideration for publication in *Nature Communications*.

Note: please check figure 4Cd, the corresponding zirconium product was missing in equation.

Our Response: We thank R1 for all the valuable comments and constructive suggestions throughout the entire revision process, which have significantly contributed to enhancing the quality of our work. We are delighted to learn that R1 is satisfied with our revisions and recommends publication in *Nature Communications*. In response to the "Note" raised by R1, we have added the corresponding zirconium product in the equation of original Figure 4Cd (Figure 5Cd in the revised manuscript).

Reviewer 2 (R2):

Recommendation: Reviewer thanks the authors for addressing all comments, in some cases even doing additional experiments. In the present text this has led to satisfactory extra information. This excellent paper can be advised for publication without modifications.

Our Response: We thank R2 for all the valuable comments and constructive suggestions throughout the entire revision process, which have significantly contributed to enhancing the quality of our work.

Reviewer 3 (R3):

Recommendation: The mechanistic issues I previously raised have been adequately addressed by the authors, but I still have some issues with the way the introduction is written.

For additional examples of stoichiometric Zr-mediated C-C bond cleavage, Bull. Chem. Soc. Jpn. 1999, 72, 2591 by Takahashi ("Carbon-Carbon Bond Cleavage and Selective Transformations of Zirconacycles") and references therein should be cited, and the strategy of reductive cleavage of zirconacycles should be mentioned in the main text.

For catalytic alkane C-C bond cleavage and alkane metathesis by immobilized Zr catalysts, the work of Basset and coworkers should be mentioned: *Acc. Chem. Res.* 2010, 43, 323 and *Angew. Chem. Int. Ed.* 1998, 37, 806.

Other than this, the zirconocene hydride product for Figure 4Cd is missing.

Some copy editing for idiomatic usage of English should be done, but otherwise, the manuscript is suitable for publication.

Our Response: We thank R3 for all the valuable comments and constructive suggestions throughout the entire revision process, which have significantly contributed to enhancing the quality of our work. We have carefully revised the introduction part of our manuscript based on R3's suggestions. Please see the following detailed points-to-points responses.

Comments and Response:

1. For additional examples of stoichiometric Zr-mediated C-C bond cleavage, *Bull. Chem. Soc. Jpn.* **1999**, 72, 2591 by Takahashi ("Carbon-Carbon Bond Cleavage and Selective Transformations of Zirconacycles") and references therein should be cited, and the strategy of reductive cleavage of zirconacycles should be mentioned in the main text.

For catalytic alkane C-C bond cleavage and alkane metathesis by immobilized Zr catalysts, the work of Basset and coworkers should be mentioned: *Acc. Chem. Res.* **2010**, 43, 323 and *Angew. Chem. Int. Ed.* **1998**, 37, 806.

Our Response: We thank R3 for the suggested information and references. We have discussed the zirconacycle work from Takahashi and the immobilized Zr catalysts from Basset in the revised manuscript as follows with 8 related references cited: "It is also worth mentioning that since the 1990s, Negishi, Takahashi, and Xi have studied the chemistry of zirconacycles, the transformation of which with other unsaturated molecules usually involved a β , β' -C-C bond cleavage.⁷⁷⁻⁸¹ All the former instances used (over) stoichiometric amounts of zirconium, and no precedents of catalytic methods using homogeneous zirconium catalysis had been developed—to the best of our knowledge (The use of heterogeneous zirconium catalysis for C-C bond cleavage was reported by Basset⁸²⁻⁸⁴)."

Here references 77-84 are:

77. Takahashi, T. *et al.* Selective skeletal rearrangement by carbon-carbon bond activation. *J. Chem. Soc., Chem. Commun.* 182-183 (1990).

78. Takahashi, T., Kageyama, M., Denisov, V., Hara, R. & Negishi, E. Facile cleavage of the C_{β} - $C_{\beta'}$ bond of zirconacyclopentenes. Convenient method for selectively coupling alkynes with alkynes, nitriles, and aldehydes. *Tetrahedron Lett.* **34**, 687-690 (1993).

79. Takahashi, T. *et al.* Selective Intermolecular Coupling of Alkynes with Nitriles and Ketones via β , β' -Carbon-Carbon Bond Cleavage of Zirconacyclopentenes. *J. Org. Chem.* **63**, 6802-6806 (1998).

80. Takahashi, T., Kitora, M., Hara, R. & Xi, Z. Carbon–Carbon Bond Cleavage and Selective Transformation of Zirconacycles. *Bull. Chem. Soc. Jpn.* **72**, 2591-2602 (2005).
81. Takahashi, T. *et al.* Zirconium Mediated Regioselective Carbon-Carbon Bond Formation Reactions. *Chem. Lett.* **21**, 331-334 (2006).
82. Corker, J. *et al.* Catalytic Cleavage of the C-H and C-C Bonds of Alkanes by Surface Organometallic Chemistry: An EXAFS and IR Characterization of a Zr-H Catalyst. *Science* **271**, 966-969 (1996).
83. Dufaud, V. & Basset, J.-M. Catalytic Hydrogenolysis at Low Temperature and Pressure of Polyethylene and Polypropylene to Diesels or Lower Alkanes by a Zirconium Hydride Supported on Silica-Alumina: A Step Toward Polyolefin Degradation by the Microscopic Reverse of Ziegler–Natta Polymerization. *Angew. Chem. Int. Ed.* **37**, 806-810 (1998).
84. Basset, J.-M., Coperet, C., Soulivong, D., Taoufik, M. & Cazat, J. T. Metathesis of Alkanes and Related Reactions. *Acc. Chem. Res.* **43**, 323-334 (2010).

2. The zirconocene hydride product for Figure 4Cd is missing.

Our Response: We thank R3 for pointing this out, we have added the corresponding zirconium product in the equation of Figure 4Cd (Figure 5Cd in the revised manuscript).